# Wafer Surface Defect Detection Based on Background Subtraction and Faster R-CNN

**DOI:** 10.3390/mi14050905

**Published:** 2023-04-23

**Authors:** Jiebing Zheng, Tao Zhang

**Affiliations:** 1School of Computer Science and Technology, Soochow University, Suzhou 215006, China; 2School of Computer Science and Engineering, Changshu Institute of Technology, Suzhou 215500, China

**Keywords:** defect detection, background subtraction, period measurement, image reconstruction, Faster R-CNN

## Abstract

Concerning the problem that wafer surface defects are easily confused with the background and are difficult to detect, a new detection method for wafer surface defects based on background subtraction and Faster R-CNN is proposed. First, an improved spectral analysis method is proposed to measure the period of the image, and the substructure image can then be obtained on the basis of the period. Then, a local template matching method is adopted to position the substructure image, thereby reconstructing the background image. Then, the interference of the background can be eliminated by an image difference operation. Finally, the difference image is input into an improved Faster R-CNN network for detection. The proposed method has been validated on a self-developed wafer dataset and compared with other detectors. The experimental results show that compared with the original Faster R-CNN, the proposed method increases the mAP effectively by 5.2%, which can meet the requirements of intelligent manufacturing and high detection accuracy.

## 1. Introduction

The manufacturing process for semiconductors includes many complex technical processes. Some improper operations may cause damage to wafer products. As an important link in the manufacturing process, defect detection aims to judge the types of defects and analyze the causes of defects to timely adjust the equipment, correct the operation, and avoid causing immense losses [1]. Defect detection was performed manually before, which was not only costly but also inefficient and no longer met the needs in the development of the modern industry.

Wafer surface defect detection has gone through three generations: (1) image processing-based algorithms that align the template image with the wafer image and highlight the defect area by difference operation [2,3,4,5]; (2) machine learning (ML)-based algorithms that utilize the machine learning algorithm to classify the defect area [6,7,8]; (3) deep learning-based algorithms that apply a deep convolutional neural network for classification and localization [9,10,11,12,13].

In the first stage based on image processing, wafer products are first converted into digital signals by an imaging system, then the template image is aligned with the wafer image to filter background information, and finally the defect areas are classified based on manually designed features. The detection methods based on image processing require the preparation of template images. In the actual production activities, it is difficult to obtain the template image. Moreover, different types of wafer products require the corresponding template image, and there are often differences between the template image and the wafer image in brightness, angle, scale, and other aspects, which can easily cause false defects.

Machine learning algorithms, such as support vector machine (SVM) and decision tree (DT), can automatically learn the mapping relationship between features and results, and can efficiently complete classification without manual design of classification criteria. The detection method based on machine learning firstly applies image processing algorithms to extract defect areas and features, and then feeds these features to ML algorithms for classification. This kind of method improves the classification part, but still requires image processing algorithms to extract features manually.

In recent years, deep learning algorithms have been used in wafer surface defect detection. Deep learning algorithms can automatically extract image features and complete classification and localization, and have a high accuracy. Once the deep learning model is built, detection personnel only need to input wafer images into the model, without complex image processing steps. The detection method based on deep learning can reduce the difficulty of algorithm development, and it has high detection performance, but this kind of method requires a lot of image data to learn the distribution of the defects.

With the fast development of the semiconductor industry, the chip structure is becoming smaller, and the wafer surface texture is becoming increasingly complex. This brings great challenges to wafer surface defect detection. As shown in Figure 1, part of the wafer surface defect overlaps with the background, and the gray values of both are close, which makes the defect easily confused with the background. In addition, some defects are small and easily lead to missed detection. It is difficult to apply the traditional methods based on image processing technology to wafer surface defect detection. In this paper, an improved detection method based on background subtraction and Faster R-CNN (Region based Convolutional Neural Network) [14] is proposed. The main contributions of this work are as follows:

(1) A background subtraction algorithm is proposed to eliminate the interference of the background. The algorithm is divided into three parts: period measurement, image reconstruction, and image difference. The difference image obtained by the background subtraction algorithm can be utilized for detection [15], segmentation [16], and other tasks [17].

(2) An improved Faster R-CNN network is proposed to detect the defects in the difference image. The spatial attention module [18] is adopted to enhance the feature extraction ability of the deep network and to improve the detection performance of large defects. In addition, the anchor boxes are initialized by the K-means algorithm to make the detector more suitable for predicting wafer surface defects.

## 2. Wafer Surface Defect Detection Method

### 2.1. Solution Overview

Figure 2 shows the whole process of the proposed method. First, frequency spectrum analysis is used to conduct the period measurement of input images, and the substructure image is obtained. Taking the substructure image as the template, local template matching is performed on input images. Then, the background image is reconstructed on the basis of the matching results and the substructure image. Image difference processing is performed on the input image and reconstructed image to generate a difference image, which is used as an input to the Faster R-CNN network. The features of the image are extracted through the backbone network and the feature pyramid network [19]. Finally, classification and position regression are carried out based on the features.

### 2.2. Background Subtraction

#### 2.2.1. Period Measurement

The common texture period measurement methods include the spectral analysis method, autocorrelation method, and gray-level co-occurrence matrix method. Regarding the image as a 2D signal, the spectral analysis method positions the peak value and extracts the texture period in the frequency-domain spectrum. However, this method is vulnerable to the interference of defect components and secondary texture elements, and so it is difficult to accurately measure the texture period. With the autocorrelation method, an autocorrelation function is used to describe the autocorrelation degree of an image in two directions, namely row and column. Period measurement is then performed in accordance with the position where the peak value of the autocorrelation function appears. This method requires the image to have good autocorrelation; otherwise, it is easy to have a certain range of deviation. The gray-level co-occurrence matrix describes the image texture information through the probability that a certain gray-level structure appears repeatedly in the image. This method is used to obtain the spatial correlation of the image by extracting its texture structure. Nevertheless, there are some problems, such as a large amount of computation and frequent loss of information, which lead to inaccurate period calculation.

This paper proposes an improved spectral analysis method. It can accurately measure the image period and is not affected by small-scale defects. Taking Figure 3a as an example, 2D discrete Fourier transform [20] is performed on the image. The process is described below:(1)$$F(u,v)=\sum_{x=0}^{H-1}\sum_{y=0}^{W-1}f(x,y)\cdot \exp[-2\pi (\frac{ux}{H}+\frac{vy}{W})j],$$
where *W* and *H* refer to the width and height of the image, respectively; *f*(*x*,*y*) refers to the original image; and *F*(*u*,*v*) refers to the spectral image. The spectral image of Figure 3a is shown in Figure 3b. It describes the frequency and amplitude distribution information of the image texture. Specifically, there is a relationship between the spacing of periodic peaks in the spatial domain, and the spacing of peaks in the frequency domain, as shown below:(2)$$\Delta u=\frac{W}{\Delta x},$$
(3)$$\Delta v=\frac{H}{\Delta y},$$where Δ*x* refers to the period in the direction of the row in the original image, Δ*y* refers to the period in the direction of the column in the original image, Δ*u* refers to the spacing of the peaks in the direction of the row in the spectral image, and Δ*v* refers to the spacing of the peaks in the direction of the column in the spectral image. The peak value of the Fourier spectrum will be affected by the existence of noise, defect, and secondary texture, and directly calculating the peak spacing will lead to inaccurate period calculation. Therefore, this study selects the areas near the central column and central row in the spectral image to make gray projection, thereby forming two 1D vectors:(4)$$Hori=\{mean(F_{H/2-2,j},F_{H/2-1,j},F_{H/2,j},F_{H/2+1,j},F_{H/2+2,j})\},\;j=0,1,\ldots ,W-1,$$
(5)$$Vert=\{mean(F_{i,W/2-2},F_{i,W/2-1},F_{i,W/2},F_{i,W/2+1},F_{i,W/2+2})\},\;i=0,1,\ldots ,H-1.$$

Then, 1D discrete Fourier transform [21] is performed on the two 1D vectors:(6)$$F(u\prime )=\sum_{n=0}^{W-1}Hori(n)\cdot \exp(-2\pi \frac{u\prime n}{W}j),$$
(7)$$F(v\prime )=\sum_{n=0}^{H-1}Vert(n)\cdot \exp(-2\pi \frac{v\prime n}{H}j),$$
where *F*(*u’*) and *F*(*v’*) refer to the frequency spectrum sequence of the 1D vectors. Figure 3c shows the frequency spectrum sequence signals of the *Hori* vector projected from the red box in Figure 3b. On the basis of the Fourier transform, the relationship between the peak spacing (Δ*u*, Δ*v*) in the 2D spectrum and the peak spacing (Δ*u’*, Δ*v’*) in the 1D spectral signal can be obtained:(8)$$\Delta u\prime =\frac{W}{\Delta u},$$
(9)$$\Delta v\prime =\frac{H}{\Delta v}.$$

From Formulas (2), (3), (8), and (9), the peak spacing (Δ*u’*, Δ*v’*) of the 1D spectral signal is almost equal to the period (Δ*x*, Δ*y*) of the original image. The most common method for obtaining the peak value is to perform a secondary derivative on the sequence signal. However, defects and secondary texture elements will cause interference peaks. Before performing a secondary derivative on the 1D spectral signal, it is necessary to apply a filter to eliminate outliers. After the main peak values are obtained, the average value of the spacing between peaks is extracted as the image period.

#### 2.2.2. Reconstruction of Background Image

The background image can be reconstructed on the basis of the period features of the wafer image. The period features of the wafer image refer to the periodic structural parameters of the image, including texture number, texture period, and texture uniformity. Among them, texture period is equal to the period of the image (Δ*x*, Δ*y*); the number of textures distributed horizontally is *n_x_* = floor(*W*/Δ*x*); the number of textures distributed longitudinally is *n_y_* = floor(*H*/Δ*y*). The wafer image can be divided into multiple repeated substructure images by texture period and texture number. Through the random sampling and superimposed averaging of substructure images, the approximate defect-free substructure image can be obtained. Then, a complete wafer image can be reconstructed through tiling the substructure image in accordance with the texture period and the texture number. However, this operation requires high accuracy of the texture period and accurate positioning of all substructure images. Otherwise, a large number of false defects will be generated in the defect extraction stage. Therefore, the template matching method is required to position and calibrate the substructure image. Nevertheless, the traditional template matching method has some problems, such as dislocated matching and low accuracy. A local template matching method in the mode of sliding window, which can effectively solve the problem with dislocated matching and improve the matching accuracy, is proposed in this paper. The substructure image is taken as the matching template, and the sliding window is set to be the matching region. The sliding window should include a complete substructure image, but it should not be too large to avoid redundant calculations. In this study, the width of the sliding window is set to be 1.2 times of the width of the period, and the height of the window is 1.2 times the height of the period. The horizontal sliding distance of the sliding window is set to a single time of the period width, and the longitudinal sliding distance is set to a single time of the period height. The sliding window starts to move from the upper left of the image to be measured. The matching template just needs to match in the sliding window.

Figure 4 shows the schematics of different template matching algorithms. Figure 4a provides a schematic of the traditional matching algorithms. The template keeps moving in the image and calculates the similarity, with a large matching area. However, the matching targets are arranged periodically in the period image. There is only one target to be matched in one local area. Therefore, there are more redundant calculations in the matching of period images by traditional matching algorithms. The improved matching algorithm in this paper is shown in Figure 4b. The image to be matched is divided into many matching areas by the sliding window. The template image only needs to be matched in the local range, which greatly reduces the irrelevant matching area and thus improves the phenomenon of dislocated matching.

To improve the matching accuracy, the Sobel edge detection algorithm is used to extract the template image and shape information in the area to be matched, and to perform matching and positioning on the basis of shape features, taking cosine similarity [22] as the measurement criterion:(10)$$S(x,y)=\frac{p_{x,y}\cdot T}{\left||p_{x,y}||\times ||T|\right|},$$
where *T* refers to the shape feature vector of the template image; *p*_*x*,*y*_ refers to the shape feature vector of the area to be matched; *S*(*x*,*y*) refers to the similarity between the template image and the area to be matched. The matching point with the highest similarity in the sliding window is selected to compare with the given threshold. If it is greater than the given threshold, then this point will be added to the matching point set. The threshold value determines the positioning accuracy of the matching, and directly affects the quality of the reconstructed image. Given an extremely low threshold value, the poor matching result will be tolerated, which leads to a substantial substructure image offset and further produces false defects. Figure 5 shows the matching scores of the sample image. It can be seen from the scores that images without defects have a high matching score, with a minimum value of 0.909702. Therefore, the threshold should not be less than 0.9. It is set to 0.9 in this study. On the basis of the coordinates in the matching point set, multiple substructure images can be extracted from the wafer image. Then, they are superposed and averaged to further obtain the substructure image with accurate registration. Owing to the interference of defects, the matching results may be missing in some areas. In this study, the least square method is used to fit point sets to obtain the fitted lines in the two directions, namely row and column. The intersection of the two lines can be used as the matching result in the missing area. Finally, the substructure image is tiled in accordance with the period features and matching coordinate point sets to reconstruct the background image for subsequent processing.

#### 2.2.3. Image Difference

Image difference can detect the different information between two images, which is usually used in anomaly detection in fixed scenes, moving object detection, and defect detection on product surfaces. This method requires two images with the same size, and the differential operation is performed in accordance with pixel points:(11)R(x,y)= I(x,y)−T(x,y)+128,
where *I*(*x*,*y*) refers to the image to be measured, *T*(*x*,*y*) refers to the background image, and *R*(*x*,*y*) refers to the difference image. Figure 6a is taken as an example. On the basis of the matched coordinate information, the background image is reconstructed, as presented in Figure 6b. Subsequently, the differential operation is performed between the input image and the background image to generate the difference image, as shown in Figure 6c. It can be seen from the difference image that the complex background information is eliminated, and the defect information is completely retained.

### 2.3. Faster R-CNN Network

Once the background subtraction operation had been completed, the following step in the process was the detection of wafer surface defects. The Faster R-CNN network was used as the detection algorithm. The reason for this is that the Faster R-CNN network, as a representative of the two-stage object detection framework, has excellent detection performance, especially in small object detection.

The overall framework of the Faster R-CNN network is shown in Figure 7. The network structure consists of three modules, which are backbone, neck, and head. Before the image is fed into the backbone module, it needs to be preprocessed. The preprocessing operations include scaling, random horizontal flipping, and random vertical flipping. The scale operation zooms the image to 1472 × 1472 pixels, which is more suitable for model computing processing. The flip operation is performed to increase the variety of image data.

The backbone network is an important part of the Faster R-CNN, which is responsible for extracting the features of the input image. In order to improve the feature representation ability of the model, ResNet50 [23] was selected as the backbone network in this study. ResNet50 is composed of four residual modules, which consist of a different number of residual blocks. The residual block uses a shortcut connection and element-wise addition to perform the residual function without an extra parameter. In Figure 7, ‘Conv 1 × 1, s1’ represents a convolution operation with scale 1 × 1 and step 1. To improve the detection performance of large defects, this paper applied two attention blocks in ResNet50.

As a neck network, the feature pyramid network (FPN) was used to process multi-scale features. The multi-scale feature maps of the backbone network are used as the input of the FPN. The feature maps are first convolved by 1×1 kernel to unify the number of channels into the same dimension. Then, the high-level feature map is transformed to the scale of the next layer map through the up-sampling operation, and the feature fusion between two maps is completed by adding corresponding elements. Finally, in order to reduce the aliasing effect caused by the up-sampling operation, 3×3 convolution is performed on the fused feature maps to generate the final feature map. The shallow feature maps are suitable for small-object detection, while the deep feature maps are favorable for detecting large objects.

The head network generates candidate boxes that may contain defects by region proposal network (RPN), and uses the RoI pooling method to extract the corresponding feature maps for prediction. Then, the feature maps are flattened into one-dimensional vectors and extracted further for classification and regression by two fully connected (FC) layers. The output of the Faster R-CNN network contains predicted-box width and height, center-point coordinates, confidence level, and the probability of each category.

The size of wafer surface defects varies greatly. The size of particle defects may be less than 36 × 36 pixels, while there are also large area defects such as stains. The preset size of the anchor boxes cannot satisfy the scale distribution of defects. In this study, the K-means algorithm was used to cluster the annotated data, and the clustering results were used to initialize the size of the anchor boxes. This can provide the model with prior knowledge, improve the learning efficiency of the model, and accelerate the convergence rate.

To improve the detection accuracy of large-scale defects, a spatial attention module was introduced into the residual block, named attention block. The spatial attention module computes the inter-spatial relationship between various spatial positions on the image and captures long-distance relationships, which are particularly effective for large-scale defects. The spatial attention module used in this study is shown in the red box in Figure 7. Firstly, the feature maps are applied by average-pooling and max-pooling operations along the channel axis. Then, the two processed maps are concatenated, and a convolution operation with a kernel size of 7 × 7 is applied to generate a spatial attention map. Finally, the former feature maps are multiplied by the spatial attention map to focus on informative parts. The mathematical expression of the spatial attention map is as follows [18]:M_s_(F) = σ(*f*^7×7^([*AvgPool*(F); *MaxPool*(F)])),(12)
where F represents the feature maps; *AvgPool* and *MaxPool* are average-pooling and max-pooling operations, respectively; *f*^7×7^ represents a convolution operation with a kernel size of 7 × 7; σ denotes the sigmoid function; M_s_ represents the spatial attention map. Deep feature maps containing rich semantic information are used to detect large-scale defects; so, the attention block was applied on the deep feature maps in this study.

## 3. Experimental Results and Analysis

### 3.1. Experimental Environment and Dataset

In this study, C++ and OpenCV (4.5.4 version) were adopted to complete the background subtraction section. Python (3.7) and PyTorch (1.7.0) were used as the deep learning frameworks to build the Faster R-CNN network. The operation system was Ubuntu 16.04.7. The GPU was NVIDIA Tesla P100 PCIE 16G. During the Faster R-CNN network training, the number of model training rounds (epochs) was 300, and the batch size was 8. Stochastic gradient descent was selected as the model parameter optimizer; the initial learning rate was 0.005, the momentum factor was 0.9, and the weight attenuation co-efficient was 0.0005. The learning rate of each parameter group decayed by 0.5 every 50 epochs.

The dataset was derived from the data collected by a wafer manufacturer and was manually labeled by experts. The dataset contained four types of defects, namely particles, scratches, ripples, and stains, with a total of 1179 defects. Figure 8 shows a part of the defect samples. Figure 8a shows the particle defects, which are caused by dust in the air that adheres to the wafer surface, or by the impacts of external sharp objects. The particles are mostly round and have a small area. Figure 8b shows the scratch defect, which is long, thin, and discontinuous. The scratches are usually caused by improper instrument operation. Figure 8c shows a ripple defect, which is characterized by wavy edges and low contrast with the background. Figure 8d shows a stain defect, which is obvious and has large area. The stains are caused by the residual dirt of the etching solution. The dataset consists of 551 images, each with a resolution of 1500 × 1500 pixels. The dataset was divided into a training dataset and test dataset in a ratio of 4:1. The defect data distribution of the wafer dataset is shown in Table 1.

### 3.2. Detection Results

To verify the reliability and superiority of the proposed method, various wafer surface defect detection methods were compared. This study adopted average precision (AP) and mean average precision (mAP) as the evaluation criteria. AP is the average of precision on the P–R curve, which was used to measure the detection accuracy of a single category. mAP is the mean value of the average precision for each category. All evaluation indexes are calculated when the threshold of intersection over union (IoU) is equal to 0.5. The defect detection algorithms for comparison include the proposed method, RetinaNet [24], Faster R-CNN [14], Sparse R-CNN [25], and YOLOv7 [26].

The qualitative detection results of the proposed method are shown in Figure 9. Figure 9a shows the manually annotated images, and Figure 9b shows the difference images output by the background subtraction algorithm. It can be seen from the difference images that the background information of wafer images was eliminated and the defect information was retained, but there were also some false defects caused by mismatching. Figure 9c shows the detection results of the proposed method. The label above the bounding box represents the predicted category and confidence level. In the case of tiny defects, the proposed method detected more defect results compared to other detectors. The reason is that the background information was eliminated by the proposed background subtraction algorithm, and then the detector focused on the defect areas. In the top row of Figure 9c, there is a particle bounding box with a 50% confidence level. The reason for the low confidence level is that the particle defect is narrow and nearly rectangular, unlike regular round particles. This problem can be improved by increasing defect diversity through data enhancement such as rescale. As for the large defects such as stains, compared with Faster R-CNN detector, the proposed method has high positioning accuracy due to the spatial attention module.

Table 2 shows the quantitative detection results of the detection methods. From the indexes mentioned above, it can be seen that the proposed method outperforms all comparative models with the highest mAP (82.7%), which is 10.0%, 5.2%, 8.2%, and 7.8% higher than RetinaNet, Faster R-CNN, Sparse R-CNN, and YOLOv7, respectively. The results show that the proposed method can detect most of the wafer surface defects and provide accurate location information, which can meet the requirements of intelligent manufacturing and high detection accuracy.

### 3.3. Ablation Study

To verify the effectiveness of the background subtraction algorithm and the spatial attention module, ablation experiments were conducted on the method in this study, and the results thereof are shown in Table 3. Compared with the original Faster R-CNN, the addition of the background subtraction algorithm improved the mAP by 3.4%. The result means that the background subtraction algorithm can effectively eliminate background interference and improve the detection performance. The addition of the spatial attention module alone to Faster R-CNN improved the mAP by 1.7%. For large area defects such as stains, the spatial attention module brought a mAP increase of 1.8%. When two optimization methods were used simultaneously, the final detection accuracy reached 82.7%, which is 5.2% higher than the detection accuracy of the original Faster-RCNN algorithm. This finding reveals that the detection performance can be better when two optimization methods are combined.

### 3.4. Effectiveness Analysis of the Period Measurement Algorithm

To verify the effectiveness of the proposed period measurement algorithm, it was compared with the spectral analysis method and the autocorrelation method through experiments. The period measurement experiment was carried out with the images that had different textures in the dataset to verify the effectiveness of the proposed algorithm with varying degrees of texture interference. The various types of wafer texture are shown in Figure 10. There are three types of wafer products with different textures, of which texture types 1 and 2 are complex, and texture type 3 is simple.

Table 4 shows the experimental results of various period measurement algorithms. In Table 4, the upper part of the figure refers to the width of the image element; the lower part of the figure refers to the height of the image element. The calculation results of the spectral analysis method and autocorrelation methods are not stable, in which some values are greatly different from the results measured manually. By contrast, the method proposed in this paper can calculate more accurate period results, with an average deviation of 0.366 pixels. The spectral analysis method has certain robustness against complex texture interference. Nevertheless, when the size of the image element is relatively small, the density of the element becomes high, which will cause a certain interference in the spectral peak extraction, resulting in an inaccurate period measurement. The autocorrelation method has a good period measurement effect on images with simple texture, but it can be easily interfered with by secondary texture. On the contrary, the method proposed in this paper can be applied to various images with complex texture, showing stable calculation results. It also has good robustness to defect interference.

### 3.5. Performance Analysis of Training Dataset

Defect detection methods based on deep learning require a large number of annotated data for training, but, in actual scenarios, it is often difficult to obtain high-quality annotated data. In order to verify the performance of the proposed method with a small number of training data, the size of the training dataset was changed and the performance of the method using the same test dataset was compared. Table 5 shows the test results of the proposed method using different numbers of training data. With the decrease in training data, the overall detection performance of the model decreases. In the case of particles, the AP values decrease by 3.3% because the size and shape of particles have little impact. However, scratches and stains vary widely in size, shape, and orientation, which makes model training difficult. Therefore, the number of scratches and stains has a great impact on the model’s performance. As for ripples, the size and shape vary more widely than in particles, so the AP value decreases by 10.2%. The results show that the number of training data has an impact on the model’s performance, especially regarding the defects with a large shape and size difference.

## 4. Conclusions

In order to reduce background interference in wafer surface defect detection and improve detection accuracy, an improved detection method based on background subtraction and Faster R-CNN is proposed in this paper.

To eliminate the background information of a wafer image, the proposed background subtraction algorithm measures the image period by spectral analysis and reconstructs the background image. The difference in operation between the wafer image and the reconstructed image can effectively eliminate the complex background information of the wafer image and highlight the defect area. The algorithm can also be applied as a preprocessing method for image annotation, image segmentation, and other tasks.

This paper applies a Faster R-CNN model to detect the defects in wafer images after background subtraction. To improve the detection performance of large defects, a spatial attention module is used to improve the Faster R-CNN structure. The experimental results showed that the performance of the proposed method is better than with other defection methods. Compared with the original Faster R-CNN algorithm, the proposed method increased mAP by 5.2%.

The disadvantage of the proposed method is that the background subtraction algorithm requires a long processing time and a high precision period. Our future work will simplify the background subtraction algorithm to reduce the time taken while keeping the effect the same.

## Figures and Tables

**Figure 1 micromachines-14-00905-f001:**
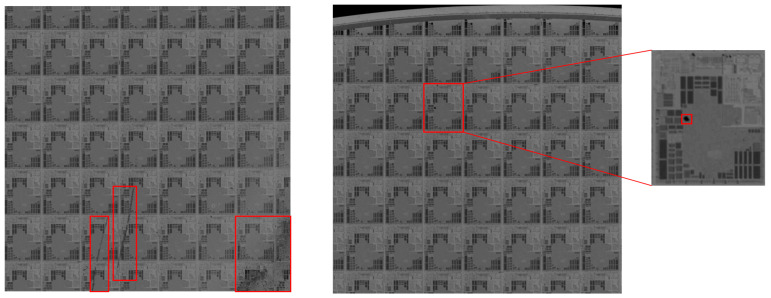
Defects on the surface of a wafer.

**Figure 2 micromachines-14-00905-f002:**
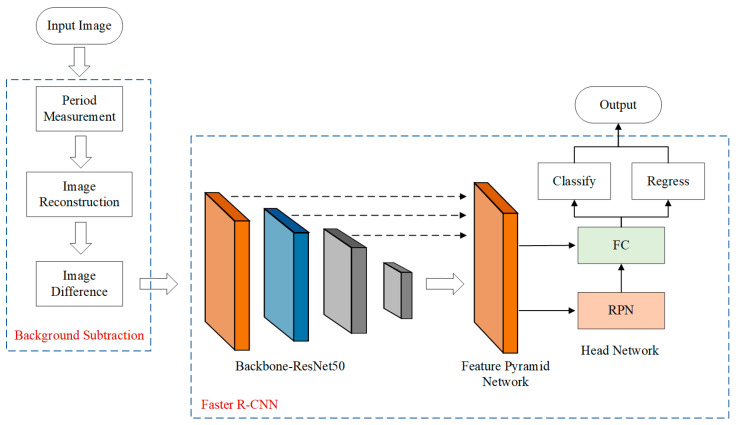
Flow process diagram of the proposed method.

**Figure 3 micromachines-14-00905-f003:**
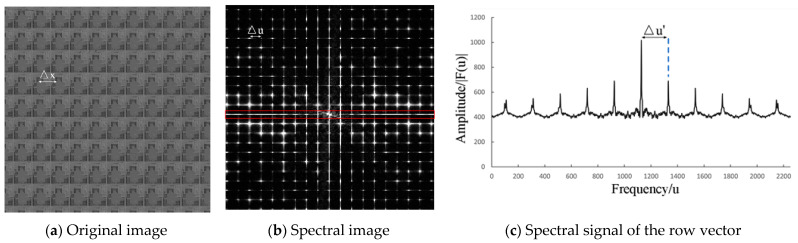
Spectrum analysis result of the removal algorithm.

**Figure 4 micromachines-14-00905-f004:**
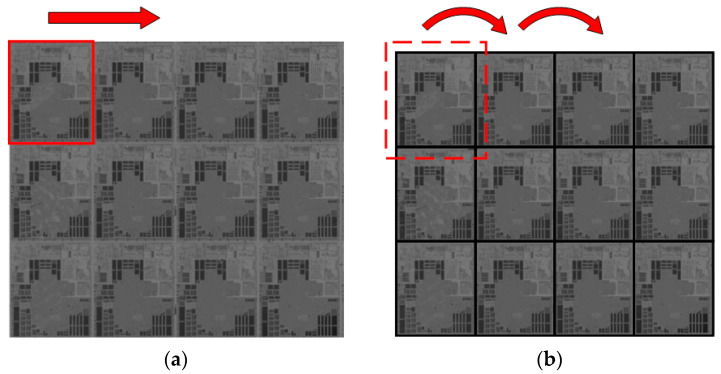
Schematics of template matching algorithms: (**a**) traditional matching algorithm. The red box represents the template image matching area; (**b**) local template matching algorithm. The red dotted box represents the sliding window.

**Figure 5 micromachines-14-00905-f005:**
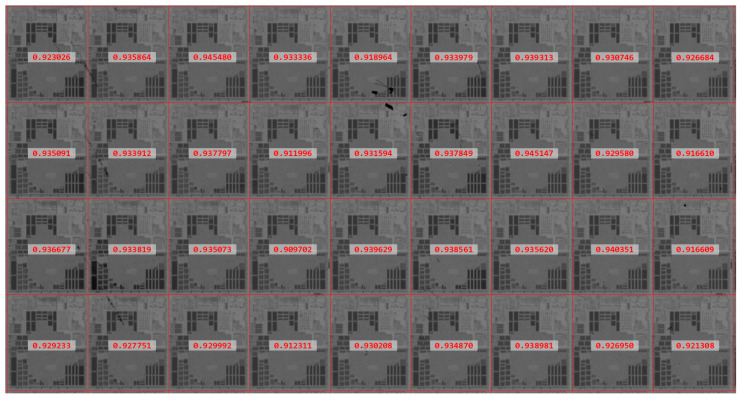
Scores of the template matching.

**Figure 6 micromachines-14-00905-f006:**
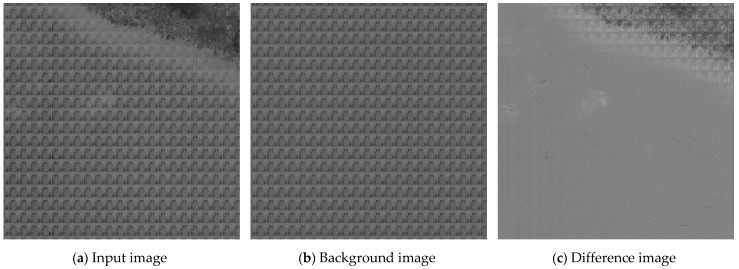
Diagram of the image difference process.

**Figure 7 micromachines-14-00905-f007:**
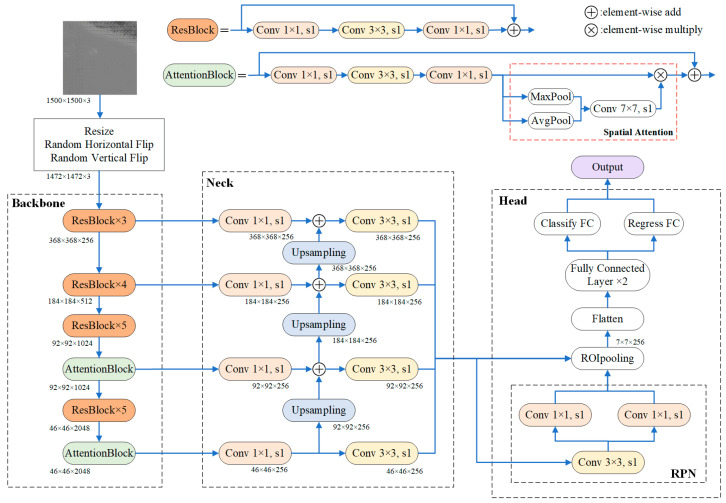
Faster R-CNN network framework.

**Figure 8 micromachines-14-00905-f008:**
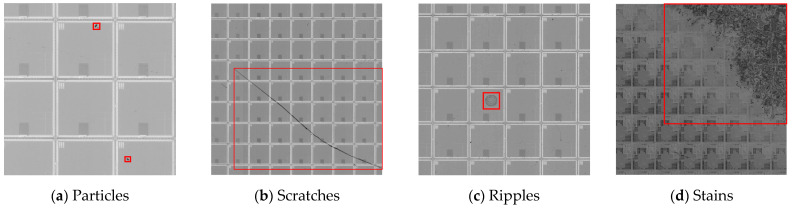
Examples of wafer surface defect.

**Figure 9 micromachines-14-00905-f009:**
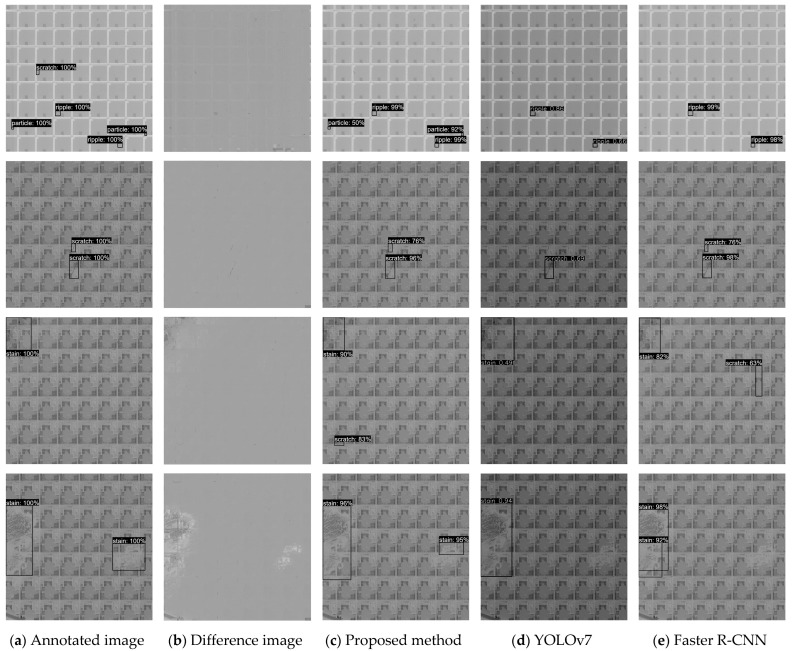
Qualitative detection results of the detection methods.

**Figure 10 micromachines-14-00905-f010:**
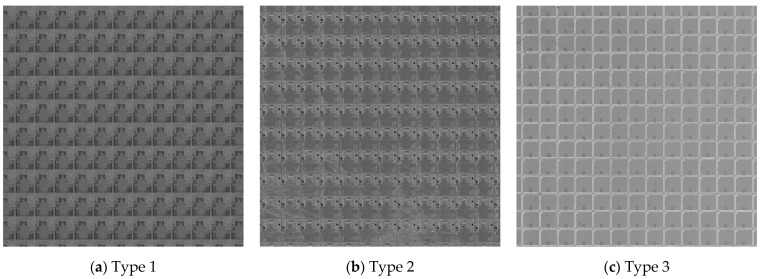
Wafers with different texture types.

**Table 1 micromachines-14-00905-t001:** Defect data distribution of the wafer dataset.

	Particles	Scratches	Ripples	Stains
Training Dataset	284	323	196	164
Test Dataset	64	67	47	34

**Table 2 micromachines-14-00905-t002:** Quantitative detection results of the detection methods.

Model	Backbone Network	AP/%				mAP/%
		Particles	Scratches	Ripples	Stains	
RetinaNet	ResNet50	76.6	63.8	79.3	70.9	72.7
Faster R-CNN	ResNet50	81.0	68.8	84.6	75.5	77.5
Sparse R-CNN	ResNet50	79.6	66.1	79.2	73.2	74.5
YOLOv7	CBS + ELAN	73.4	67.3	85.2	73.9	74.9
Proposed Method	ResNet50 + Attention	85.7	75.3	92.2	77.7	82.7

**Table 3 micromachines-14-00905-t003:** Results of the ablation study.

Model	Background Subtraction	Spatial Attention	AP/%				mAP/%
			Particles	Scratches	Ripples	Stains	
Faster R-CNN			81.0	68.8	84.6	75.5	77.5
	√		83.9	71.3	89.2	76.2	80.2
		√	81.5	70.0	88.0	77.3	79.2
	√	√	85.7	75.3	92.2	77.7	82.7

**Table 4 micromachines-14-00905-t004:** Comparison of results by period measurement algorithms.

Texture Type	Manual Measurement	Spectral Analysis	Autocorrelation	Proposed Method
Type 1	204	204.5	208.0	203.6
	243	243.6	218.0	242.6
Type 2	200	200.5	200.0	200.0
	240	239.1	159.7	239.3
Type 3	185	373.5	184.7	184.5
	183	182.8	184.0	182.8

**Table 5 micromachines-14-00905-t005:** Results from different numbers of training data.

Model	Numbers of Training Data				AP/%				mAP/%
	Particles	Scratches	Ripples	Stains	Particles	Scratches	Ripples	Stains	
Proposed Method	284	323	196	164	85.7	75.3	92.2	77.7	82.7
	253	287	169	140	85.2	63.5	86.2	67.6	75.6
	221	249	146	118	84.5	58.4	88.5	70.6	75.5
	171	198	118	99	82.1	52.5	82.6	60.2	69.3
	141	163	95	79	82.4	46.1	82.0	62.5	68.3

## Data Availability

The data presented in this study are available on request from the corresponding authors.

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
