# Peer review of "Wafer Surface Defect Detection Based on Background Subtraction and Faster R-CNN"

_micromachines, 2023, doi:10.3390/mi14050905_

Round 1

Reviewer 1 Report

Comments are attached.

Author Response

Thank you very much for your comments and suggestions. Those comments are all valuable and very helpful for revising and improving our paper, as well as the important guiding significance to our researches. We have studied comments carefully and have made correction which we hope meet with approval. The main corrections in the paper and the responds to your comments are as following:

Comment 1: Introduction is not adequate to present the significance and importance of this work. For instance, after introducing the need for the defect detection of wafer, the article directly starts with the present remarkable works, then lists some typical establishments on the detection methodology of wafer defects in the past years. In fact, as detection method, authors should have outlined the process of the developments of the wafer detection from the first proposal, how many stages of improvements in technologies and performance, in which what key specifications were solved and what key technologies or strategies were developed.

Response 1: We have re-written the introduction part according to your suggestion. We have outlined the process of the developments of the wafer detection and introduced each stage in detail.

Comment 2: In line 63, R-CNN is the first time to be used, so the full-name of this abbreviation - “Region based convolutional neural network” should be given.

Response 2: Thanks for your suggestion. We have added the full-name of R-CNN.

Comment 3: Figure 1 is used to present the easy confusion of the surface defects with the background subtraction, so more efficient descriptions and explanations are needed.

Response 3: As suggested by the reviewer, we have added more descriptions to explain Figure 1 on line 61.

Comment 4: On line 163, Figure 3 should be Figure 4.

Response 4: Thanks for your careful checks. we have made the corrections on line 163 & 169.

Reviewer 2 Report

1. Add references for all the equations in this manuscript.

2. Provide more explaination for the wafer surface defect in Figure 7.

3. Of course supplement more information for the Faster R-CNN network framework. 244

4. Why the threshold was set to be lower than 0.9, it is hard to follow.

  • 5. Conclusions are suggested to be stated in terms of points.

6. Provide the limitation of your work.

  •  

Author Response

Thank you very much for your comments and suggestions. Those comments are all valuable and very helpful for revising and improving our paper, as well as the important guiding significance to our researches. We have studied comments carefully and have made correction which we hope meet with approval. The main corrections in the paper and the responds to your comments are as following:

Comment 1: Add references for all the equations in this manuscript.

Response 1: As suggested by the reviewer, we have added references for the equation we referenced. The rest equations are the derivation equations of this paper.

Comment 2: Provide more explanation for the wafer surface defect in Figure 7.

Response 2: According to your comment, we have added explanation for the wafer surface defect on line 403. We have described the characteristics and causes of the defects.

Comment 3: Of course supplement more information for the Faster R-CNN network framework.

Response 3: We have modified Figure 7 (Faster R-CNN network framework) to present more information and supplemented more information for the Faster R-CNN network framework.

Comment 4: Why the threshold was set to be lower than 0.9, it is hard to follow.

Response 4: Thanks for your comment. We have added our experimental result to support this conclusion. Experimental result shows that image without defects have a high matching score, and the minimum value is greater than 0.9.

Comment 5: Conclusions are suggested to be stated in terms of points.

Response 5: We have re-written this part in terms of points according to your suggestion.

Comment 6: Provide the limitation of your work.

Response 6: We have provided the limitation of our work on line 541. Specific limitations are as follows: 1.our method takes much processing time; 2.the reconstruction of the background image requires a high precision period.

Reviewer 3 Report

In this work, the authors propose an improved detection method based on background subtraction and Faster R-CNN for defect detection. I find this manuscript to be well written and interesting, although there are some issues the authors should address prior to consideration for publication, see below.

·        Paragraph starting on line 163: Presumably the authors meant the discussion to focus on Fig. 4, not Fig. 3 as written in the text.

·        Can the authors discuss how the method performs with smaller training datasets? For the 4 defect types in Table 1, >80% of the defects were used in the training dataset.

·        Can the authors discuss the confidence level of 50% for particles with the proposed method (Fig. 8c, top row), and what steps can be taken to improve that?

Author Response

Thank you very much for your comments and suggestions. Those comments are all valuable and very helpful for revising and improving our paper, as well as the important guiding significance to our researches. We have studied comments carefully and have made correction which we hope meet with approval. The main corrections in the paper and the responds to your comments are as following:

Comment 1: Paragraph starting on line 163: Presumably the authors meant the discussion to focus on Fig. 4, not Fig. 3 as written in the text.

Response 1: Thanks for your careful checks. we have made the corrections.

Comment 2: Can the authors discuss how the method performs with smaller training datasets? For the 4 defect types in Table 1, >80% of the defects were used in the training dataset.

Response 2: As suggested by the reviewer, we have supplemented the experiment in which the model was trained under different number of training data and discussed the results in section 3.5. The results show that the number of training data has an impact on the model performance, especially the defects with large shape and size difference.

Comment 3: Can the authors discuss the confidence level of 50% for particles with the proposed method (Fig. 8c, top row), and what steps can be taken to improve that?

Response 3: According to your comments, we have discussed the case (the confidence level of 50% for particles) and given a solution to improve the case on line 434. The reason for the low confidence level is that the particle defect is narrow and nearly rectangular, unlike regular round particles. This problem can be improved by increasing defect diversity through data enhancement such as rescale.